# Plasmacytoid Dendritic Cell (pDC) Infiltration Correlate with Tumor Infiltrating Lymphocytes, Cancer Immunity, and Better Survival in Triple Negative Breast Cancer (TNBC) More Strongly than Conventional Dendritic Cell (cDC)

**DOI:** 10.3390/cancers12113342

**Published:** 2020-11-12

**Authors:** Masanori Oshi, Stephanie Newman, Yoshihisa Tokumaru, Li Yan, Ryusei Matsuyama, Pawel Kalinski, Itaru Endo, Kazuaki Takabe

**Affiliations:** 1Department of Surgical Oncology, Roswell Park Comprehensive Cancer Center, Buffalo, NY 14263, USA; Masanori.Oshi@RoswellPark.org (M.O.); snewman5@buffalo.edu (S.N.); Yoshihisa.Tokumaru@roswellpark.org (Y.T.); 2Department of Gastroenterological Surgery, Yokohama City University Graduate School of Medicine, Yokohama 236-0004, Japan; ryusei@yokohama-cu.ac.jp (R.M.); endoit@yokohama-cu.ac.jp (I.E.); 3Department of Surgery, Jacobs School of Medicine and Biomedical Sciences, State University of New York, Buffalo, NY 14263, USA; 4Department of Surgical Oncology, Graduate School of Medicine, Gifu University, 1-1 Yanagido, Gifu 501-1194, Japan; 5Department of Biostatistics & Bioinformatics, Roswell Park Comprehensive Cancer Center, Buffalo, NY 14263, USA; li.yan@roswellpark.org; 6Department of Medicine, Roswell Park Comprehensive Cancer Center, Buffalo, NY 14263, USA; Pawel.kalinski@roswellpark.org; 7Division of Digestive and General Surgery, Niigata University Graduate School of Medical and Dental Sciences, Niigata 951-8520, Japan; 8Department of Breast Surgery, Fukushima Medical University School of Medicine, Fukushima 960-1295, Japan; 9Department of Breast Surgery and Oncology, Tokyo Medical University, Tokyo 160-8402, Japan

**Keywords:** biomarker, breast cancer, conventional DC, dendritic cell, gene expression, survival, plasmacytoid DC, TNBC, treatment response, xCell

## Abstract

**Simple Summary:**

Dendritic cells (DC) represent a major antigen-presenting cell in the tumor immune microenvironment (TIME) and play an essential role in cancer immunity. Clinical relevance of plasmacytoid DC (pDC) and conventional DC (cDC) infiltration were studied in large patient cohorts using transcriptomic analyses. We found that high pDC was significantly associated with better survival in triple negative breast cancer (TNBC), but not cDC. High pDC TNBC tumors enriched immune and inflammation-related gene sets including IFN-γ signaling more strongly than cDC TNBC, which also enriched metastasis-related gene sets. pDC correlated with CD8^+^ and CD4^+^ memory T cells, and had a cytolytic activity score stronger than cDC in TNBC. High pDC TNBC were associated with a high fraction of anti-cancer immune cells and high expression of all the immune checkpoint molecules examined. For the first time, we found that pDC are correlated with immune response and survival in TNBC patients more strongly than cDC.

**Abstract:**

Dendritic cells (DC) represent a major antigen-presenting cell type in the tumor immune microenvironment (TIME) and play an essential role in cancer immunity. Conventional DC (cDC) and plasmacytoid DC (pDC) were defined by the xCell algorithm and a total of 2968 breast cancer patients (TCGA and METABRIC) were analyzed. We found that triple-negative breast cancer (TNBC) had a high fraction of cDC and pDC compared to the other subtypes. In contrast to cDC, high pDC in TNBC was significantly associated with better disease-specific and disease-free survival consistently in both cohorts. High cDC TNBC tumors enriched not only inflammation and immune-related, but also metastasis-related gene sets in Gene Set Enrichment Analysis, whereas high pDC TNBC enriched inflammation and immune -related gene sets including IFN-γ signaling more strongly than cDC. pDC TNBC correlated with CD8^+^, CD4^+^ memory, IFN-γ score, and cytolytic activity stronger than cDC TNBC. High pDC TNBC were associated with a high fraction of anti-cancer immune cells and high expression of all the immune check point molecules examined. In conclusion, pDC levels correlated with the infiltration of immune cells and patient survival in TNBC more strongly than cDC; this is the first study suggesting the clinical relevance of pDC infiltration in TNBC.

## 1. Introduction

Infiltration of immune cells into the tumor microenvironment (TME) cause host–tumor interaction and plays a significant role in cancer progression and treatment response, thus now known to be a prognostic factor in cancer [1,2,3]. The tumor immune microenvironment (TIME) is a subject of particular interest in triple-negative breast cancer (TNBC), known to have the most abundant immune cell infiltration despite being the most aggressive subtype in breast cancer [4,5,6]. 

Dendritic cells (DC) are a major group of antigen-presenting cell among innate immune cells and play an essential role in cancer immunity [7]. DCs process antigens and present them to T cells in response to pathogens and cancer cells, serving as a link between the innate and the adaptive immune responses [8]. In addition to carrying antigens from the tumor to the lymph nodes, thus initiating tumor-specific immunity, DCs also maintain anti-tumor effects by reactivating T cells at the tumor sites [9,10]. 

DCs are divided into conventional DCs (cDCs) and plasmacytoid DCs (pDCs) [11]. cDCs present cancer antigens to T cells, thus playing a major role in the induction of cancer immunity. pDCs, although existing at similar numbers as cDC, have a low antigen uptake capacity and little was known in its role in anti-tumor immunity until recently [12,13]. pDCs are reported to express major histocompatibility complex (MHC) class II and costimulatory molecules, and can function as antigen presenting cells [14] and as immunomodulatory cells producing large amounts of type I interferon (IFN) in response to single-stranded viral RNA and DNA. pDC infiltration to solid tumors is reported to have negative or positive impacts on antitumor immune response depending on the contest [15,16,17,18]. Given these controversial data, it was of interest to investigate the clinical relevance of pDC infiltration in breast cancer patients.

Tumor immune microenvironment (TIME) has been analyzed traditionally by flow cytometry or immunohistochemistry [19]. However, these approaches in the clinical setting require a significant number of tumor samples and labor-intensive analyses for accurate quantification. Recently, a number of computational algorithms that quantify immune cells using tens to hundreds of cell marker gene expressions have been introduced [20,21,22,23]. One such algorithm, xCell, allows for the estimation of 64 immune and stromal cell types from gene expression signatures from either gene expression microarray or mRNA-sequencing [21]. Our group has been utilizing computational biological analyses on the transcriptome of bulk tumors to elucidate the clinical relevance of TIME in breast cancer [24,25,26,27,28]. Here, we used the transcriptomic markers and methodology of xCell to compare the roles of cDCs and pDCs in the TIME of breast cancer patients by using the Cancer Genome Atlas (TCGA, *n* = 1065) and Molecular Taxonomy of Breast Cancer International Consortium (METABRIC, *n* = 1903) cohorts. 

## 2. Results

### 2.1. Triple-Negative Breast Cancer (TNBC) Has High Fraction of Conventional Dendritic Cell (cDC) and Plasmacytoid Dendritic Cell (pDC)

Conventional dendritic cells (cDCs) and plasmacytoid dendritic cells (pDCs) were defined using the gene expression profiles by the xCell algorithm [21], which is also listed in Appendix A. TNBC was significantly associated with a consistently high pDC and cDC fraction in both TCGA and METABRIC cohorts (Figure 1B, Appendix A; all *p* < 0.001). These results suggest that TNBC has high infiltration of both cDCs and pDCs. 

### 2.2. High pDC Fraction, but Not cDC, Was Significantly Associated with Better Survival in TNBC, but Not Other Forms of Breast Cancer

Since both cDCs and pDCs were highly infiltrated in TNBC, we chose to further investigate the clinical relevance of these DCs in this subtype. Given that DCs are antigen presenting cells that activate the immune cell cycle and cancer immunity, high DC TNBC was expected to associate with better survival. The median was used to divide patients into low and high groups in each cohort. First, we examined the association of abundance of cDCs or pDCs with clinical outcomes. In the whole cohort, the abundance of cDCs or pDCs was not associated with disease specific survival (DSS) in either of the TCGA or METABRIC cohorts (Figure 1 and Appendix A). Interestingly, although the high cDC TNBC was not associated with DSS nor disease-free survival (DFS) in either of the cohorts (Figure 2A), a high pDC TNBC was significantly associated with better DSS and DFS in the TCGA cohort, and better DSS in the METABRIC cohort (Figure 2B). 

In contrast to TNBC, neither cDCs nor pDCs were associated with DSS in the whole cohort of the TCGA (Figure 1) and METABRIC (Appendix A). 

### 2.3. TNBC with High pDC Fraction Enriched Multiple Inflammation- and Immune-Related Gene Sets Stronger than cDCs

To elucidate the mechanism of why there was a survival difference in pDCs and not in cDCs, we performed the gene set enrichment analysis (GSEA) of Hallmark gene sets in both the TCGA and METABRIC cohorts. Interestingly, the high cDC TNBC enriched not only the inflammation- and immune-related gene sets; Allograft rejection, IL2-STAT5 signaling, IL6/JAK/STAT3, and tumor necrosis factor (*TNF*)-α signaling via NFkB, and also metastatic-related gene sets; transforming growth factor (*TGF*)-β, epithelial mesenchymal transition (EMT), and angiogenesis consistently in both cohorts (Figure 3A, normalized enrichment score (NES) = 1.65, 1.65, 1.61, 1.59, 1.68, 1.59, 1.55 for the TCGA, NES = 1.65, 1.59, 1.62, 1.47, 1.38, 1.37, 1.25 for the METABRIC, respectively). 

Conversely, high pDC TNBC had a higher level of NES compared to cDCs in Allograft rejection, IL2 STAT5 signaling, IL6/JAK/STAT3, inflammatory response as well as complement and IFN-γ gene sets, all related with immune response and inflammation (Figure 3B, NES = 2.04, 2.16, 2.00, 2.03, 2.19, 1.98 in the TCGA, NES = 1.83, 1.74, 1.85, 1.81, 1.91, 1.75 in the MTABRIC, respectively). These results implicate that high infiltration of pDCs in TNBC is associated with stronger inflammation and immune response compared with cDCs, which is associated with not only a weaker immune response, but also with pathways related to metastasis. 

### 2.4. High cDC and High pDC TNBC Are Infiltrated with Multiple Types of Anti-Tumor Immune Cells

Given the enrichment of immune-related gene sets, it was of interest to determine which immune cells were infiltrated in high DC TNBC. High cDC TNBC were significantly associated with a high fraction of anti-cancer immune cells; CD8^+^ T cells, CD4^+^ memory T cells, and B cells, and a low fraction of T helper 1 (Th1) in the TCGA cohort (Figure 4A; *p* = 0.038, *p* < 0.001, and *p* = 0.004, respectively). These results were validated by the METABRIC cohort, which additionally showed an increased fraction of M1 macrophages. No pro-cancer immune cells, T helper 2 (Th2), Regulatory T cells (Treg), and M2 macrophages were elevated in high cDC TNBC in the TCGA cohort. In the METABRIC cohort, high cDC TNBC was significantly associated with high infiltration of Th2 and M2 macrophages (*p* = 0.016 and *p* = 0.001, respectively). 

High pDC TNBC was also significantly associated with a high fraction of anti-cancer immune cells, CD8^+^ T cells, CD4^+^ memory T cells, M1 macrophages and B cells (Figure 4B. all *p* < 0.001). These results were consistent with the METABRIC cohort (Figure 4B. all *p* < 0.001). Similar to cDCs, the high pDC group was also associated with significantly higher Th1 in the METABRIC cohort (*p* < 0.001). No pro-cancer immune cell was elevated in high pDC TNBC of the TCGA cohort. However, in the METABRIC cohort, high pDC TNBC was significantly associated with a high fraction of Th2 and Treg (both *p* < 0.001). These results suggest that both high cDC and pDC TNBC tumors were consistently infiltrated with anti-cancer immune cells. 

### 2.5. pDC Strongly Correlated with CD8^+^ and CD4^+^ Memory T Cells, IFN-γ Score, and Cytolytic Activity (CYT) Score in TNBC

Next, we investigated the correlation of the DC fraction with major anti-cancer immune cells, CD8^+^ and CD4^+^ memory T cells as well as immune function scores, IFN-γ, and CYT scores in the TCGA and METABRIC cohorts. The CD8^+^ and CD4^+^ memory T cell fraction were measured by the xCell algorithm. The IFN-γ pathway score was calculated by the gene set variation analysis algorithm with the Hallmark gene set, similar to our previous works [29,30,31]. CYT was calculated based on the expression of granzyme A (*GZMA*) and perforin (*PRF1*), as we have previously reported [32,33,34]. cDCs showed none to very weak correlation with CD8^+^ T cells, CD4^+^ memory T cells, IFN-γ, and CYT in TNBC of both cohorts (Figure 5A; *r* = 0.188, 0.338, 0.215, 0.283 in TCGA, *r* = 0.431, 0.514, 0.353, 0.416 in the METABRIC, respectively). In contrast, pDCs showed a strong correlation with CD8^+^, CD4^+^ memory T cells, IFN- γ, and CYT in TNBC of both cohorts (Figure 5B; *r* = 0.577, 0.705, 0.749, 0.771 in TCGA, *r* = 0.727, 0.825, 0.746, 0.830 in the METABRIC, respectively). These results suggest that the pDC infiltration in TNBC strongly correlated with the infiltration of anti-cancer immune cells and cytolytic activity. 

### 2.6. High Fraction of pDC Was Associated with Uniformly Elevated Expressions of Immune Checkpoint Molecules

It has been previously reported that infiltration of DCs are increased in a tumor with enhanced anticancer immunity and increased efficacy of checkpoint blockade [35,36]. Thus, we investigated the relationship between the DC fraction and expression of major immune checkpoint molecules in both the TCGA and METABRIC cohorts. Only four out of nine immune checkpoint molecules examined were consistently elevated with high cDC TNBC in both cohorts; *PD-1*, *PD-L2*, *CTLA-4*, and *TIGIT* (Figure 6A; *p* = 0.029, <0.001, 0.002, and <0.001 for the TCGA, and *p* = 0.031, <0.001, <0.001, and <0.001 for the METABRIC, respectively). On the other hand, the high pDC TNBC group was significantly associated with the high expression of all nine immune checkpoint molecules examined: *PD-1, PD-L1, PD-L2, CTLA-4, IDO1, IDO2, LAG3, HLA-A*, and *TIGIT* in both cohorts (Figure 6B). These results suggest that the high pDC fraction TNBC may possibly respond to immune checkpoint inhibitors.

## 3. Discussion

Using the TCGA and METABRIC cohorts of breast cancer patients, we found that transcriptionally defined levels of cDC and pDC infiltration were elevated in TNBC compared to other subtypes of breast cancer. In contrast to high cDC infiltration, high pDC TNBC was significantly associated with favorable DSS and DFS. High cDC TNBC enriched not only immune and inflammation-related gene sets, but also metastasis-related gene sets, whereas high pDC TNBC had enriched immune and inflammation-related gene sets including IFN-γ more strongly than cDC. Both high cDC TNBC and high pDC TNBC were associated with anti-cancer immune cells, however, it was more consistent and robust with high pDC TNBC. We further found that pDC in TNBC was strongly positively correlated with the fraction of CD8^+^ T cells and CD4^+^ memory T cells, and the level of IFN-γ score as well as cytolytic activity score, which suggests an overall anti-cancer immune microenvironment and immune activity. Interestingly, high pDC TNBC was significantly associated with the high expression of all the immune checkpoint molecules examined, the number of which is much larger than high cDC TNBC. 

It was reported that the pDC gene signature is associated with a positive patient prognosis in lung adenocarcinoma, although the correlation was even stronger for cDCs [37]. Our current study indicates that the association of pDCs with clinical outcomes is context dependent, and, at least in breast cancer, subtype specific. We found that elevated pDCs were associated with better survival in TNBC patients, but not in the other subtypes of breast cancer, which may be due to the fact that TNBC shows an elevated immune cell infiltration [38,39] and may contain more antigenic targets. Some studies showed that pDCs can play an immune-suppressive role and facilitate cancer progression in both animal models and humans [15,16,17]. In contrast, the other studies indicated that activated tumor-associated pDCs caused tumor regression in mice [15,16,17,18]. In this study, both high pDC and cDC TNBC tumors were enriched with immune-related gene sets, while several anti-cancerous immune cells were highly infiltrated, especially in high pDC TNBC tumors. In addition, the high correlation of immune activity was suggested as the reason why high pDC was involved in the better prognosis of TNBC patients. 

Recently, immune checkpoint inhibitors (ICIs) have become a new modality against TNBC [40], however, their efficacy and appropriate patient selection remain significant challenges. DCs play a unique role in the transportation of tumor antigen and the activation of T cells, a crucial step in the response to ICI therapy [35,41]. Tumor-resident DCs regulate and maintain T cell activation during therapy [42,43]. Thus, DCs are an obvious and essential player in the anti-tumor T cell response and may be a viable therapeutic target to improve T cell response [44]. For example, pDC agonists such as imiquimod and R848 can activate and maintain immune response and improve tumor control [45,46]. Our study demonstrates that there is upregulation of key checkpoint markers including *PD-1* and *PD-L1* in high pDC TNBC as well as others. Given our results, we speculate that pDC levels may have a utility as a supportive predictive biomarker of response to ICIs in TNBC patients. 

Our study is not without limitations. We defined cDCs, pDCs, and other immune cells by the transcriptomic prolife determined by the xCell algorithm, which may or may not capture all the cells defined by the gold standard, since not all gene expression is translated. Additionally, this is a retrospective study that utilized a large amount of clinical and genetic data, however, data on co-morbidity and therapeutic intervention are missing. Finally, the biggest limitation is that our results are based on the analyses of tumor gene expression alone without any direct quantification of tumoral DCs with in vivo and vitro experiments. To this end, the results of this study should be validated in future experiments using in vitro and in vivo techniques to better understand the cause and causal relationship as well as the underlying mechanisms. 

In conclusion, transcriptionally defined high cDC or pDC TNBC were associated with a favorable immune microenvironment. High pDC TNBC was strongly correlated with high immune function compared to cDCs. This is the first study suggesting that the pDC level may be more clinically relevant than cDCs in TNBC patients. pDC level may serve as a useful tool for identifying patients with better prognosis. 

## 4. Materials and Methods

### 4.1. TCGA and METABRIC Breast Cancer Cohorts and Their Data

Clinical information and genomic profiling of The Cancer Genome Atlas (TCGA-BRCA: *n* = 1065) [47], who were female and had pathological diagnosis of breast cancer, and Molecular Taxonomy of Breast Cancer International Consortium (METABRIC: *n* = 1903) cohorts [48] were obtained through the cBio Cancer Genomic Portal [49]. The average value was used for genes with multiple probes, and the log_2_-transform of gene expression data were used for in all analyses. 

### 4.2. Cell Composition Fraction and Scores Related with Immune Activity

Immune cell fractions where defined using the xCell score, which was calculated using transcriptome data. Via the xCell algorithm, data were downloaded from online databases [21], as previously reported [50,51,52,53]. The cytolytic activity (CYT) was defined as the geometric mean of granzyme A and perforin 1 expression values [54]. 

### 4.3. Gene Set Expression Analyses

Gene set enrichment analyses (GSEA) software (Java version 4.0) [55] with MSigDB [56], Hallmark was used for gene set analysis, and a false discovery rate (*FDR*) of 0.25, as recommended by the GSEA software, was used for statistical significance.

### 4.4. Other

Statistical analyses and creating figures were undertaken using R (version 4.0.1) with several publicly available packages, and Microsoft Excel (version 16 for Windows, Richmond, WA, USA). Less than 0.05 was the *p* value for statistical significance. The Kaplan–Meier method with the log rank test was used for survival analysis. The one-way ANOVA or Fisher’s exact tests were used to calculate *p* values of comparisons between groups. Tukey type boxplots show the median and inter-quartile level values. 

## 5. Conclusions

Transcriptionally defined high cDC and pDC TNBC were associated with a favorable immune microenvironment. pDC levels were correlated with immune response and survival of TNBC patients more strongly than cDCs. To our knowledge, this is the first study to demonstrate that pDC level may be clinically relevant in TNBC patients. 

## Figures and Tables

**Figure 1 cancers-12-03342-f001:**
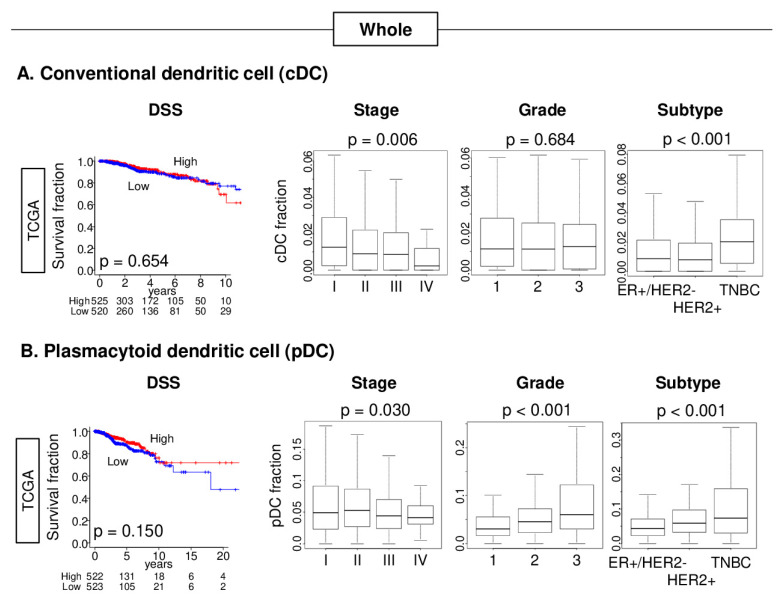
Association of cDCs and pDCs with clinical features in the TCGA cohort. Kaplan–Meier survival (disease-specific survival [DSS]) curves and boxplots of the AJCC stage, Nottingham pathological grade, and subtype with (**A**) cDCs and (**B**) pDCs. The Kaplan–Meier curve shows the DSS of cDCs and pDCs, low (blue) and high (red), with a *p*-value of the log rank test. Median was used as the cut-off to divide patients into low and high groups within each cohort. Tukey type boxplots showed median and inter-quartile level values, and the one-way ANOVA test was used to calculate *p* values.

**Figure 2 cancers-12-03342-f002:**
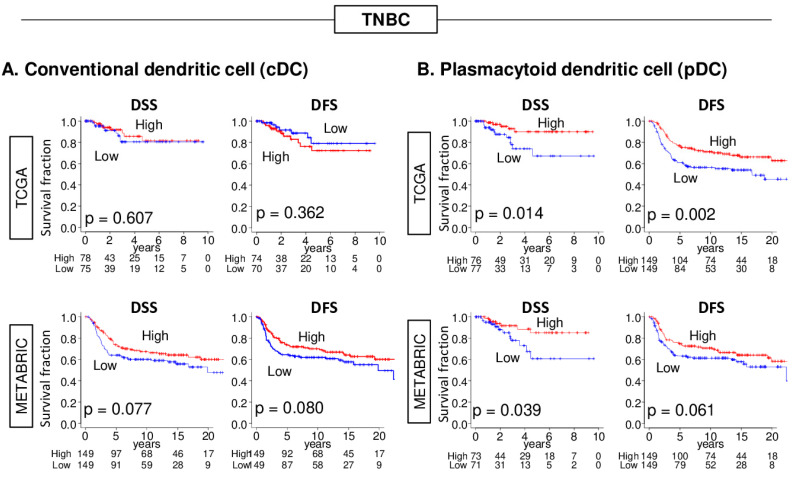
Association between cDCs or pDCs and the survival of triple-negative breast cancer (TNBC) patients in the TCGA and METABRIC cohorts. DFS and DSS of (**A**) cDC and (**B**) pDC in TNBC. Median was used as the cut-off to divide patients into low (blue) and high (red) groups within each cohort. To compare the two groups, Kaplan–Meier curves with the log-rank test was used for survival analysis.

**Figure 3 cancers-12-03342-f003:**
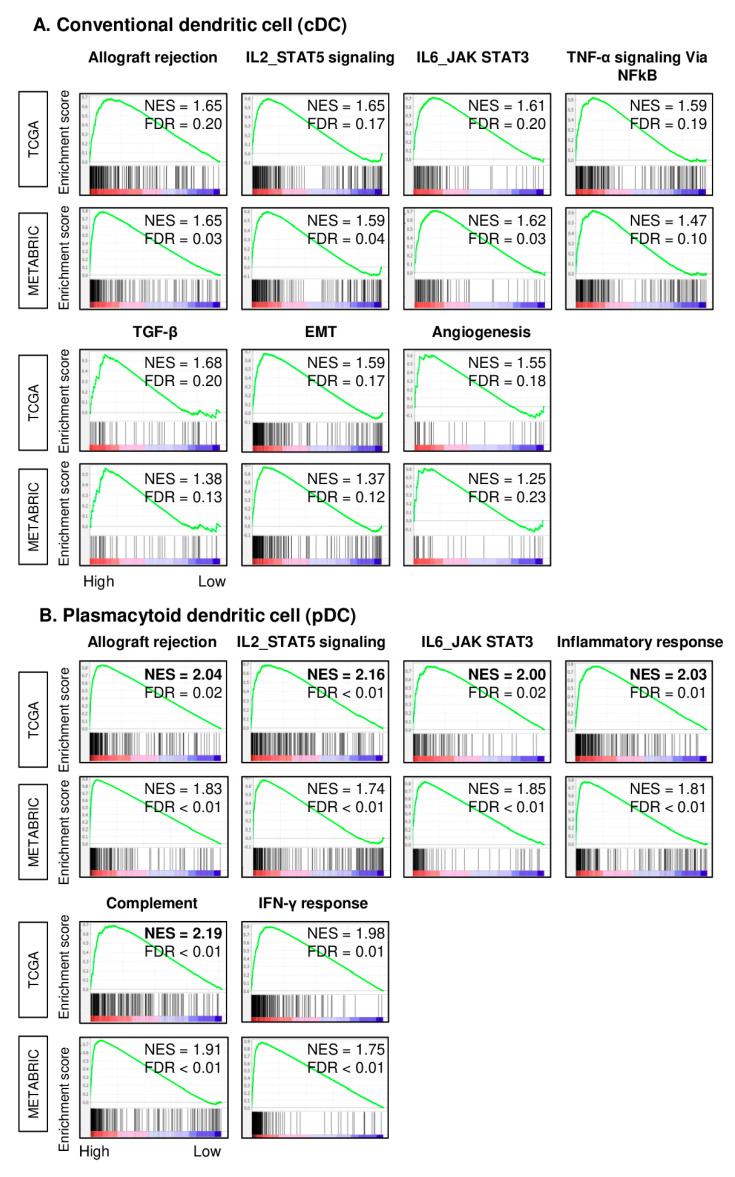
Gene set enrichment analysis (GSEA) of cDCs or pDCs in TNBC in the TCGA and METABRIC cohorts. Gene set enrichment plots of Hallmark gene sets that show significant enrichment to high (**A**) cDC or (**B**) pDC TNBC consistently in both cohorts are demonstrated. Median was used as the cut-off to divide patients into low and high groups within each cohort. NES, normalized enrichment score; FDR, false discovery rate. FDR < 0.25 is considered statistically significant.

**Figure 4 cancers-12-03342-f004:**
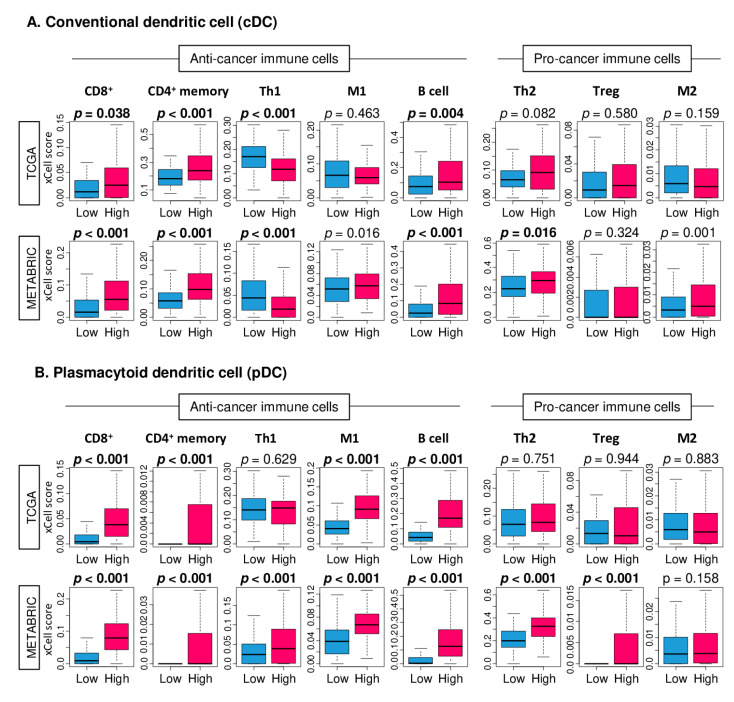
Tumor infiltrating immune cells of low and high cDC or pDC TNBC. Boxplots of the anti-cancer immune cells, CD8^+^, CD4^+^ memory, T helper type 1 cells (Th1), M1 macrophages, and B cell, and pro-cancer immune cells; T helper type 2 cells (Th2), regulatory T cell (Treg), and M2 macrophages by low and high (**A**) cDCs or (**B**) pDCs in the TCGA and METABRIC cohort. One-way ANOVA test was used to calculate the p values. Median was used as the cut-off to divide patients into low (blue) and high (high) groups within each cohort.

**Figure 5 cancers-12-03342-f005:**
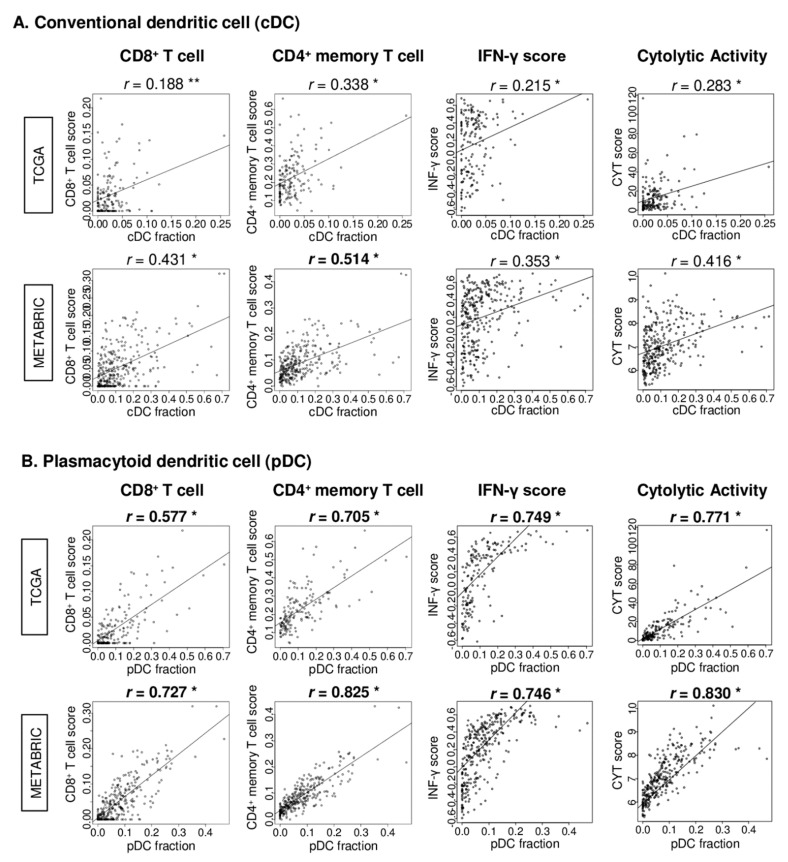
Correlation analysis of cDCs or pDCs in TNBC with CD8^+^ T cells, CD4^+^ memory T cells, IFN-γ pathway, and cytolytic activity score (CYT). Correlation plots of the (**A**) cDCs or (**B**) pDCs with CD8^+^ T cells, CD4^+^ memory T cells, IFN-γ pathway score, and CYT. *p*-value was analyzed with the Spearman *r* correlation. * *p*-value < 0.01, ** *p*-value = 0.002.

**Figure 6 cancers-12-03342-f006:**
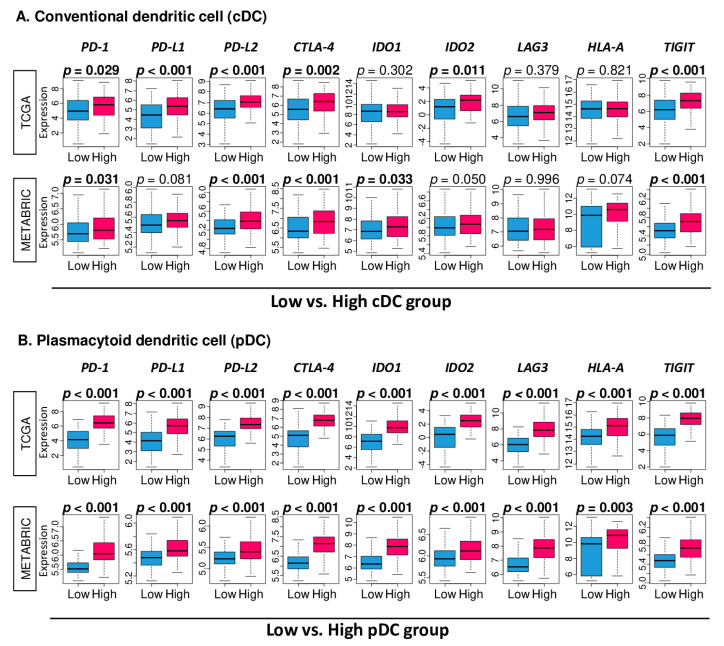
Association between the cDC or pDC and expression of immune checkpoint molecules in the TCGA and METABRIC cohorts. Blue and red boxes stand for low and high cDC or pDC groups, respectively. Gene expression levels of *PD-1*, *PD-L1* and *L2*, *CTLA4*, *IDO1* and *2*, *LAG3*, *HLA-A*, and *TIGIT* of (**A**) cDCs and (**B**) pDCs. One-way ANOVA test was used to calculate *p* values.

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
