# Peer review of "Plasmacytoid Dendritic Cell (pDC) Infiltration Correlate with Tumor Infiltrating Lymphocytes, Cancer Immunity, and Better Survival in Triple Negative Breast Cancer (TNBC) More Strongly than Conventional Dendritic Cell (cDC)"

_cancers, 2020, doi:10.3390/cancers12113342_

Round 1

Reviewer 1 Report

This report is a retrospective study on two transcriptomic databases (TCGA and METABRIC) from a total of  2968 breast cancers. A sequencing analysis using the xCell algorithm shows plasmacytoid dendritic cell (pDC) and conventional DC (cDC) infiltration particularly in Triple Negative Breast Cancers (TNBC) compared to Her2Neu or ER-positive breast cancers. It shows higher association of pDC tumor infiltration markers than cDC markers with survival and with good prognosis markers such as cytotoxic T cell infiltration and checkpoint inhibitor expression. It defines well the limits of the study (retrospective, with no information on individual treatments or comorbidities). It cites previous studies on pDC infiltration association with good prognosis in other cancers in mice, which are not frequent yet.  This is a very useful study to help understand how to stimulate pDC against cancer.

Table I is a list of genes, but it would be useful to state which markers were used to define "pDC" and "cDC".

Line 71 "pDC, which exist at much lower numbers" this is not true, please check in the literature.  "cDC1", are much rarer than cDC2, which are in the same order of magnitude as pDC. 

Also, in the discussion there is no comment at all on the mechanisms which may explain the statistical observations found in this manuscript. This would enhance the interest of the results. For instance the citation of Lemercier et al., Cancer Research 2013; 73:4629, would help comment the results much better.

Minor remarks

Line 42  more than 2968 breast cancer patients, remove "cohorts"

Line 52 "more strongly than cDC" instead of stronger than

line 73 "express MHC class II and costimulatory molecules, AND can function as antigen presenting cells " instead of THAT

line 76 TIME has been analyzed traditionally by flow cytometry or immunohistochemistry : please cite some studies.

Line 92. "TNBC was significantly associated with high pDC" please add "and cDC" fraction... .

Line 115 it seems that here ther are no DSF results, only DSS?

Line 219 "all checkpoint molecules examined, THE NUMBER OF which is much larger... 

LINE 235 "of response" rather than "respond"

Line 271 "more strongly" rather than "stronger"

Author Response

Reviewer #1 (Remarks to the Author):

This is a very useful study to help understand how to stimulate pDC against cancer.

Response:

First of all, we would like to thank Reviewer #1 for taking time and effort reviewing our manuscript. We are delighted to learn that the Reviewer found our study to be very useful.

Comment 1:

Table I is a list of genes, but it would be useful to state which markers were used to define "pDC" and "cDC".

Response 1:

We agree with the Reviewer that it will be useful to state which markers were used to define pDC and cDC in the Table S1. We have now modified the table and listed the genes in alphabetical order to clarify which marker was used for which DC, as below.

Table S1. Marker genes used to define conventional dendritic cell (cDC) and plasmacytoid dendritic cell (pDC) by xCell

cDC (38 genes)                                                                                 pDC (38 genes)

Gene Name

Gene Description

Gene Name

Gene Description

ACTR3

actin related protein 3

APOC3

apolipoprotein C3

ALCAM

activated leukocyte cell adhesion molecule

CACNB1

calcium voltage-gated channel auxiliary subunit beta 1

ALDH1A2

aldehyde dehydrogenase 1 family member A2

CCR2

C-C chemokine receptor type 2

ALOX15

arachidonate 15-lipoxygenase

CD2AP

CD2 associated protein

ANXA1

annexin A1

CELA2A

chymotrypsin like elastase 2A

CCDC88A

coiled-coil domain containing 88A

CSHL1

Chorionic somatomammotropin hormone like 1

CCL13

C-C motif chemokine ligand 13

CUX2

cut like homeobox 2

CCL17

C-C motif chemokine ligand 17

CXCL13

C-X-C motif chemokine ligand 13

CCL23

C-C motif chemokine ligand 23

CXCR3

C-X-C motif chemokine receptor 3

CCL24

C-C motif chemokine ligand 24

DNASE1L3

deoxyribonuclease 1 like 3

CD163

CD163 molecule

FKBP2

FKBP prolyl isomerase 2

CD1A

CD1a molecule

FLT3

fms related tyrosine kinase 3

CD1B

CD1b molecule

FUT7

fucosyltransferase 7

CD1C

CD1c molecule

GZMB

granzyme B

CD1E

CD1e molecule

HIST1H2BB

Histone cluster 1 H2B family member b

CD209

CD209 molecule

HPD

4-hydroxyphenylpyruvate dioxygenase

CD80

CD80 molecule

IDH3A

isocitrate dehydrogenase 3 (NAD(+)) alpha

CD86

CD86 molecule

IL3RA

interleukin 3 receptor subunit alpha

CD93

CD93 molecule

KCNA5

potassium voltage-gated channel subfamily A member 5

CLEC10A

C-type lectin domain containing 10A

KCNK10

potassium two pore domain channel subfamily K member 10

CLEC4A

C-type lectin domain family 4 member A

KCTD5

potassium channel tetramerization domain containing 5

CRH

corticotropin releasing hormone

LILRB4

leukocyte immunoglobulin like receptor B4

DBI

diazepam binding inhibitor

LRRC36

Leucine rich repeat containing 36

DNASE1L3

deoxyribonuclease 1 like 3

MAPKAPK2

mitogen-activated protein kinase-activated protein kinase 2

FCER1A

Fc fragment of IgE receptor Ia

MYBPC1

myosin binding protein C, slow type

FGL2

fibrinogen like 2

P2RY14

purinergic receptor P2Y14

FLT3

fms related tyrosine kinase 3

PTCRA

pre T cell antigen receptor alpha

GFRA2

GDNF family receptor alpha 2

RPL3L

ribosomal protein L3 like

ITGAX

integrin subunit alpha X

RUNX2

runt related transcription factor 2

KCNK13

potassium two pore domain channel subfamily K member 13

SCT

secretin

PITPNA

phosphatidylinositol transfer protein alpha

SLC12A3

solute carrier family 12 member 3

RAB7A

RAB7A, member RAS oncogene family

SLITRK3

SLIT and NTRK like family member 3

RRP1B

ribosomal RNA processing 1B

SPCS1

signal peptidase complex subunit 1

S100A10

S100 calcium binding protein A10

SPIB

Spi-B transcription factor

SLAMF8

SLAM family member 8

TACR1

tachykinin receptor 1

SSR1

signal sequence receptor subunit 1

TLR7

toll like receptor 7

TCTN3

tectonic family member 3

TSPAN13

tetraspanin 13

WDFY3

WD repeat and FYVE domain containing 3

ZNF221

Zinc finger protein 221

Comment 2:

Line 71 "pDC, which exist at much lower numbers" this is not true, please check in the literature.  "cDC1", are much rarer than cDC2, which are in the same order of magnitude as pDC.

Response 2:

We agree with the Reviewer that this statement was not true. We have changed the sentence in the revision manuscript as below.

Introduction section:

pDCs, although existing at similar numbers as cDC,  have a low antigen uptake capacity and little was known in its role in anti-tumor immunity until recently [12,13].

Comment 3:

Also, in the discussion there is no comment at all on the mechanisms which may explain the statistical observations found in this manuscript. This would enhance the interest of the results. For instance the citation of Lemercier et al., Cancer Research 2013; 73:4629, would help comment the results much better.

Response 3:

We totally agree with the Reviewer that additional comments on the mechanisms that explain our findings would strengthen our manuscript and we appreciate your suggestion of the reference. We have added the paragraph below in the discussion section.

Discussion section:

It was reported that pDC gene signature is associated with a positive patient prognosis in lung adenocarcinoma, although the correlation was even more strongly for cDC [37]. Our current study indicates that the association of pDC with clinical outcomes is context dependent, and, at least in breast cancer, subtype specific. We found that elevated pDC was associated with better survival in TNBC patients, but not in the other subtypes of breast cancer, which may be due to the fact that TNBC shows an elevated immune cell infiltration [38,39] and may contain more antigenic targets. Some studies showed that pDC can play an immune-suppressive role and facilitate cancer progression in both animal models and humans [15-17]. In contrast, the other studies indicated that activated tumor-associated pDC caused tumor regression in mice [15-18]. In this study, both high pDC and cDC TNBC tumors were enriched with immune-related gene sets, while several anti-cancerous immune cells were highly infiltrated, especially in high pDC TNBC tumors. In addition, the high correlation of immune activity was suggested as the reason why high pDC was involved in the better prognosis of TNBC patients.

Comment 4:

Minor remarks

Response 4:

We would like to thank for the Reviewer to point out these issues. We have now re-written these sentences in revision manuscript, as below.

Line 42  more than 2968 breast cancer patients, remove "cohorts"

Line42 changed to: 2968 breast cancer patient (TCGA and METABRIC)

Line 52 "more strongly than cDC" instead of stronger than

Line 52 changed to: more strongly than cDC

line 73 "express MHC class II and costimulatory molecules, AND can function as antigen presenting cells " instead of THAT

line 73 changed to: express MHC class II and costimulatory molecules, and can function as antigen presenting cells

line 76 TIME has been analyzed traditionally by flow cytometry or immunohistochemistry : please cite some studies.

line 76 changed to: TIME has been analyzed traditionally by flow cytometry or immunohistochemistry [15].

Line 92. "TNBC was significantly associated with high pDC" please add "and cDC" fraction... .

Line 92 changed to: TNBC was significantly associated with high pDC and cDC fraction consistently in both TCGA and METABRIC cohorts (Figure 1B, S1B; all p < 0.001).

Line 115 it seems that here ther are no DSF results, only DSS?

Line 115 changed to: In contrast to TNBC, neither cDC nor pDC were associated with DSS in the whole cohort of the TCGA (Figure 1) and METABRIC (Supplemental Figure S1).

Line 219 "all checkpoint molecules examined, THE NUMBER OF which is much larger...

Line 219 changed to: all the immune checkpoint molecules examined, the number of which is much larger than high cDC TNBC.

LINE 235 "of response" rather than "respond"

LINE 235 changed to: as a supportive predictive biomarker of response to ICIs in TNBC patients.

Line 271 "more strongly" rather than "stronger"

Line 271 changed to: survival of TNBC patients more strongly than cDC

Reviewer 2 Report

Dr. Masanori Oshi et al. reported the important study for investigating “plasmacytoid dendritic cell correlate with tumor infiltrating lymphocytes, cancer immunity and better survival in triple negative breast cancer stronger than conventional dendritic cell’’. The scientists have experienced from cancer patients infiltrate human neoplasms with poor prognosis. Only, the actual purpose of tumor-associated pDCs remains argumentative points. Because various studies showed that pDCs can play an immuno-suppressive role and facilitate tumor progression in both animal models and humans. In contrast, some studies indicated that the presence of activated tumor-associated pDCs results in tumor regression in mice. These studies provided completely different and controversial evidences. Although the authors offered the point view is not novel, some analytic results can provide the novel potential therapeutic strategies for malignant breast cancer, especially for triple negative breast cancer. Finally, the results of this paper are interesting but still can not elucidate the possible mechanisms for how the pDCs provide the pDCs-induced immunity in TNBC, the following concerns should be addressed:

1) Because the major analytic resources are dependent with TCGA and METABRIC breast cancer cohorts and their information. Although that can provide many “associated “information for comparing with pDCs and cDCs…Merely, it is hard to elucidate whether earlier stage TNBC induces more pads to offer high immunity response to make the cancer immunity and better survival or pDCs and CDCs competition in TNBC. Because the database just can collect the patients’ final body condition, in parliamentary procedure to clarify this section. 2) We advise the authors can use the TNBC cells cultured condition medium to treat any one DCs -cells line, e.g. CAL-1 or GEN2.2 or PMDC05 to provide the cause and causal relationship and potential mechanisms. Or so this weak point, the authors also saw and described in discussion section. But Cancers journal has a high impact for cancer study, we strongly advise that the writers can prove the in vitro or in vivo experimental evidence.

Author Response

Reviewer #2:

Comment 1:

Dr. Masanori Oshi et al. reported the important study for investigating “plasmacytoid dendritic cell correlate with tumor infiltrating lymphocytes, cancer immunity and better survival in triple negative breast cancer stronger than conventional dendritic cell’’. The scientists have experienced from cancer patients infiltrate human neoplasms with poor prognosis.

Response 1:

First of all, we would like to thank Reviewer #2 for taking time and effort reviewing our manuscript. We are delighted to learn that Reviewer found our study to be important.

Comment 2:

Only, the actual purpose of tumor-associated pDCs remains argumentative points. Because various studies showed that pDCs can play an immuno-suppressive role and facilitate tumor progression in both animal models and humans. In contrast, some studies indicated that the presence of activated tumor-associated pDCs results in tumor regression in mice. These studies provided completely different and controversial evidences. Although the authors offered the point view is not novel, some analytic results can provide the novel potential therapeutic strategies for malignant breast cancer, especially for triple negative breast cancer.

Response 2:

We would like to thank Reviewer #2 for highlighting this point. The fact that there are controversial evidences motivated us to conduct the current study. We have further clarified this point in the Discussion section as the following.

Introduction section:

pDC infiltration to solid tumors are reported to have negative or positive impact on antitumor immune response depending on the contest [15-18]. Given these controversial data, it was of interest to investigate the clinical relevance of pDC infiltration in breast cancer patients.”

Discussion section:

“Some studies showed that pDC can play an immune-suppressive role and facilitate cancer progression in both animal models and humans [15-17]. In contrast, the other studies indicated that activated tumor-associated pDC caused tumor regression in mice [15-18]. In this study, both high pDC and cDC TNBC tumors were enriched with immune-related gene sets, while several anti-cancerous immune cells were highly infiltrated, especially in high pDC TNBC tumors. In addition, the high correlation of immune activity was suggested as the reason why high pDC was involved in the better prognosis of TNBC patients.”

Comment 3:

Finally, the results of this paper are interesting but still can not elucidate the possible mechanisms for how the pDCs provide the pDCs-induced immunity in TNBC, the following concerns should be addressed:

Because the major analytic resources are dependent with TCGA and METABRIC breast cancer cohorts and their information. Although that can provide many “associated “information for comparing with pDCs and cDCs…Merely, it is hard to elucidate whether earlier stage TNBC induces more pads to offer high immunity response to make the cancer immunity and better survival or pDCs and CDCs competition in TNBC. Because the database just can collect the patients’ final body condition, in parliamentary procedure to clarify this section. We advise the authors can use the TNBC cells cultured condition medium to treat any one DCs -cells line, e.g. CAL-1 or GEN2.2 or PMDC05 to provide the cause and causal relationship and potential mechanisms. Or so this weak point, the authors also saw and described in discussion section. But Cancers journal has a high impact for cancer study, we strongly advise that the writers can prove the in vitro or in vivo experimental evidence.

Response 3:

We totally agree with Reviewer #2 that our result is solely based on the transcriptome of bulk tumors that does not elucidate mechanism. We also completely agree with the importance of elucidating the mechanism using in vivo and vitro experiment, which provide us better understanding and possibly lead to novel therapeutics. The novelty of our study is that we analyzed large sample size cohort of real human patients. Unfortunately, we do not have the capability to conduct the suggested experiments to provide the cause and causal relationship and potential mechanisms. We have added the limitation in the Discussion section, as below.

Discussion section:

Our study is not without limitations. We defined cDC, pDC and other immune cells by the transcriptomic prolife determined by the xCell algorithm, which may or may not capture all the cells defined by the gold standard, since not all gene expression are translated. Additionally, this is a retrospective study which utilized large amount of clinical and genetic data, however, data on co-morbidity and therapeutic intervention are missing. Finally, the biggest limitation is that our results are based on analyses of tumor gene expression alone without any direct quantification of tumoral DCs with in vivo and vitro experiment. To this end, the results of this study should be validated in future experiments using in vitro and in vivo techniques to better understand the cause and causal relationship as well as the underlying mechanisms.

Reviewer 3 Report

The paper is interesting, however I have some remarks:

1) English: as there are too many language mistakes, a careful revision of the text is mandatory.

2) Title: “Plasmacytoid dendritic cell (pDC) correlate with…”: maybe “Plasmacytoid dendritic cell (pDC) infiltration correlate with…”???

3) Abstract, lines 50-53: the phrase: “In conclusion, pDC levels correlated with immune response and infiltration of immune cells and patient survival in TNBC stronger than cDCs, representing the first study to demonstrate the clinical relevance of pDC infiltration in TNBC” should be modified as follows “In conclusion, pDC levels correlated with infiltration of immune cells and patient survival in TNBC stronger than cDCs; this is the first study suggesting a clinical relevance of pDC infiltration in TNBC”.

4) Discussion, lines 244-246: the phrase “This is the first study to demonstrate that pDC level may be more clinically relevant than cDC in TNBC patients. pDC level may serve as a useful tool for identifying patients who have a better survival” should be modified as follows “This is the first study suggesting that pDC level may be more clinically relevant than cDC in TNBC patients. pDC level may serve as a useful tool for identifying patients with better prognosis”

Author Response

Reviewer #3 (Remarks to the Author):  

The paper is interesting, however I have some remarks:

Response: 

First of all, we would like to thank Reviewer #3 for taking time and effort reviewing our manuscript. We are delighted to learn that the Reviewer found our manuscript to be interesting.

Comment 1:

1) English: as there are too many language mistakes, a careful revision of the text is mandatory.

Response 1:

We apologize that our use of the English language was not of publication quality. We have carefully revised and corrected the grammar throughout the entire manuscript and marked the changes using “Track Changes” function.

Comment 2:

2) Title: “Plasmacytoid dendritic cell (pDC) correlate with…”: maybe “Plasmacytoid dendritic cell (pDC) infiltration correlate with…”???

Response 2:

We agree with the Reviewer that the title was inaccurate. We have now revised the title to the following.

Title:

Plasmacytoid dendritic cell (pDC) infiltration correlate with tumor infiltrating lymphocytes, cancer immunity and better survival in triple negative breast cancer (TNBC) more strongly than conventional dendritic cell (cDC)”

Comment 3:

3) Abstract, lines 50-53: the phrase: “In conclusion, pDC levels correlated with immune response and infiltration of immune cells and patient survival in TNBC stronger than cDCs, representing the first study to demonstrate the clinical relevance of pDC infiltration in TNBC” should be modified as follows “In conclusion, pDC levels correlated with infiltration of immune cells and patient survival in TNBC stronger than cDCs; this is the first study suggesting a clinical relevance of pDC infiltration in TNBC”.

Response 3:

We agree with the Reviewer that the suggested sentence is more succinct and clearer. We have revised it as below.

Abstract section:

In conclusion, pDC levels correlated with infiltration of immune cells and patient survival in TNBC more strongly than cDC; this is the first study suggesting a clinical relevance of pDC infiltration in TNBC.

Comment 4:

4) Discussion, lines 244-246: the phrase “This is the first study to demonstrate that pDC level may be more clinically relevant than cDC in TNBC patients. pDC level may serve as a useful tool for identifying patients who have a better survival” should be modified as follows “This is the first study suggesting that pDC level may be more clinically relevant than cDC in TNBC patients. pDC level may serve as a useful tool for identifying patients with better prognosis”

Response 4:

We would like to thank the Reviewer for the suggestion. We have now re-written the sentence in revision manuscript, as below.

Discussion section:

“This is the first study suggesting that pDC level may be more clinically relevant than cDC in TNBC patients. pDC level may serve as a useful tool for identifying patients with better prognosis.”